# Design and Initial Testing of an Affordable and Accessible Smart Compression Garment to Measure Physical Activity Using Conductive Paint Stretch Sensors

**Ben Greenspan and Michele A. Lobo ***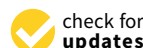

Department of Physical Therapy & Biomechanics & Movement Science Program, College of Health Sciences, University of Delaware, 540 S College Avenue, Room 210K Newark, DE 19713, USA; bengspan@udel.edu

* Correspondence: malobo@udel.edu; Tel.: 30-2-831-8526

**Abstract:** Motion capture and the measurement of physical activity are common practices in the fields of physical therapy, sports medicine, biomechanics, and kinesiology. The data collected by these systems can be very important to understand how someone is recovering or how effective various assistive devices may be. Traditional motion capture systems are very expensive and only allow for data collection to be performed in a lab environment. In our previous research, we have tested the validity of a novel stitched stretch sensor using conductive thread. This paper furthers that research by validating a smart compression garment with integrated conductive paint stretch sensors to measure movement. These sensors are very inexpensive to fabricate and, when paired with an open-sourced wireless microcontroller, can enable a more affordable, accessible, and comfortable form of motion capture. A wearable garment like the one tested in this study could allow us to understand how meaningful, functional activities are performed in a natural setting.

**Keywords:** wearable technology; stretch sensor; open-sourced; motion capture; e-textile; accessibility

## 1. Introduction

The measurement of physical activity through motion capture is very common in the fields of physical therapy and related disciplines [1,2]. To truly measure how humans are moving, how someone is recovering from an injury, or how effective an assistive device is, these systems need to measure the user across settings, not just in a lab environment. The gold standard of motion capture systems is very expensive and uses high-speed cameras positioned around the perimeter of a room that require data collections to be performed in laboratory settings [3–5]. A less expensive way to measure movement within a lab setting is through the use of the Microsoft Kinect, which utilizes both an (red, green, and blue) RGB color camera and an infrared depth sensor to determine joint angles [6]. Systems like this work well for large movements but cannot compete with the density of information gathered from one of the full-room optical systems.

Outside of a lab environment, there are three main types of systems to measure movement, which are: optical data from a smartphone, use of inertial measurement units (IMUs), or flexible on-body sensors. As smartphone cameras continue to capture images and videos with higher resolution, they can be paired with apps to measure movement. One example of this can be seen with the Angles App, which acts as a video goniometer app [7]. When a camera is not possible or patients prefer a camera not to be present, motion capture systems using inertial measurement units (IMUs) have become more common. Unfortunately, these systems require bulky sensors to be attached to the body [3,8–10]. These bulky sensors can make motion capture uncomfortable and lead to unnatural

movement [3,9–12]. Lastly, a variety of textile-based sensors are currently being researched, but none have made their way into a common commercial product [13]. Wearable sensors have been made from silver coated nylon thread [14–16], conductive knit fabrics [17], carbon nanotubes [18], dielectric elastomers [19,20], and P(VDF-TrFE) fibers [21]. All these sensors track motion via measured changes in resistance or capacitance [22–24].

Most of the above sensors can be difficult to acquire or to manufacture, requiring wet lab space, inkjet printers, or commercial screen printers to fabricate. Because our lab focuses on affordable, open-sourced technology, we have been working within the domain of conductive threads and, more recently, conductive paint for our stretch sensors. Previous testing has shown that conductive thread could be stitched to produce a stretch sensor with a large change in resistance that was repeatable over time [16]. Unfortunately, because silver can oxidize, and a thin coating can make this process happen more quickly, over time the resistance of the sensor will grow too large to read. To investigate an alternative, even-more-affordable option, this study aimed to test the validity and reliability of carbon-based conductive paint sensors integrated within commercially available compression garments to measure flexion activity at the elbow and shoulder. To our knowledge, no other studies have used conductive ink to create garment-based movement sensors. We selected to measure elbow and shoulder flexion, because movement at these joints is important for the performance of a variety of daily activities that are impaired for individuals with upper extremity disabilities. Measurement of movement at these joints is especially important to measure the effectiveness of exoskeletons and other devices that aim to improve arm function. We aimed to implement a single-case study design to test the smart garment system on an adult and child participant to determine the accuracy of scaled versions of this affordable system relative to the established, more-expensive trakSTAR motion analysis system [25].The trakSTAR is an IMU-based system and was selected over an optical system, because it provides the best accuracy outside of a lab setting.

There is a need for affordable, accessible, comfortable, and wireless forms of motion capture like the one tested in this study [26]. Data collections can track more meaningful, functional activities when they can be performed in natural, real-world environments where expensive and wired forms of motion capture are not feasible. A low-cost, wireless form of motion capture has the potential to allow for activity classification and continuous tracking of kinematics. This can support or replace existing data collection systems in the natural environment that typically rely on time-intensive coding of behavior from video. In order to document clinically relevant differences for patient populations, a wearable system should be accurate in tracking joint angles within roughly 15 degrees [27]. While 15 degrees is not near the accuracy of what a finely calibrated optical system can achieve, the benefit of constant data collection outside of a lab setting is significant and will provide new insight on how users are moving throughout the day in natural settings.

## 2. Materials and Methods

### 2.1. Fabrication and Integration of the Conductive Paint Sensors

The commercially available conductive paint (Electric Paint, Bare Conductive, UK) that we purchased is originally intended to "draw" circuits, replacing where copper wire would be for short distances, or for cold soldering for those without soldering experience. This conductive paint is affordable at only USD 34.95 per 50 mL and was specified to bind to the fabric to be used in our selected garment. The Material Safety Data Sheet for the paint can be found online [28]. The paint is comprised of water, natural resin, conductive carbon, and a humectant. The paint is unlikely to cause skin irritation and is not sensitizing to the respiratory system or skin. There is no evidence of mutagenic potential, carcinogenicity, or reproductive toxicity for this product. It should not be ingested and can cause slight irritation with eye exposure. If the conductive paint was painted onto a sheet of paper using a $\frac{1}{2}$-wide brush, the resistance would be roughly 52 ohms ($\Omega$)/inch [29]. From our previous study, it was found that a 90% polyester/10% spandex jersey knit (Polyester Jersey Knit, Denver Fabrics,

USA) was the best fabric substrate for a stretch sensor due to its advantageous mechanical recovery, so we utilized this fabric substrate for the current project as well [30–33]. Multiple methods of applying the conductive paint to the fabric were tested. The best results came from first cutting the fabric to a desired length (6 inches long by 2 inches wide, in the lengthwise or weft direction, then prestretching the fabric to the maximum length it would be stretched to when worn by the user—roughly 30% strain. By prestretching the fabric, it may be that the paint can adhere to a greater number of individual threads within the fabric while reducing mechanical strain on the paint. The fabric was clamped at this distance using metal alligator clips, and a custom 3D printed curved piece was placed underneath the fabric to further expose the individual threads. This part was printed with PLA plastic on a (Printrbot Plus, Printrbot, Lincoln, CA, USA) 3D printer. Next, the conductive paint was applied as a thin bead down the center of the fabric, and a plastic squeegee was used to make a roughly $\frac{1}{2}$-wide line of paint down the center of the fabric sensor (Figure 1). The paint was left to dry, while the fabric was still stretched for 30 min. Once the paint was dry in this configuration, the resistance was approximately 0.5 kΩ/inch. A 6 sensor where 5 was painted had an unstretched resistance of approximately 2.6 kΩ. Preliminary resistance data were measured on a multimeter (Model 9205B+, Adafruit Industries, New York, NY, USA). This sensor was cyclically elongated and retracted for 250 cycles on the same testing apparatus as in Greenspan et al.'s 2018 paper. When stretched to a 30% strain, the sensor reached roughly 3.4 kΩ (Figure 2).

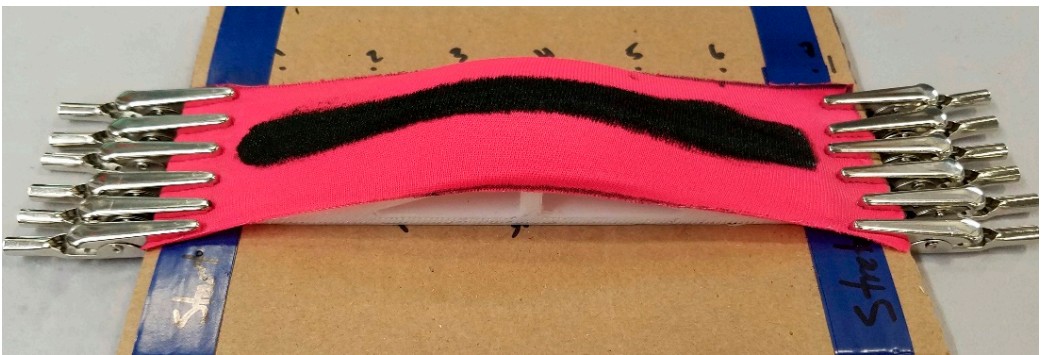

**Figure 1.** Conductive paint drying on stretched fabric.

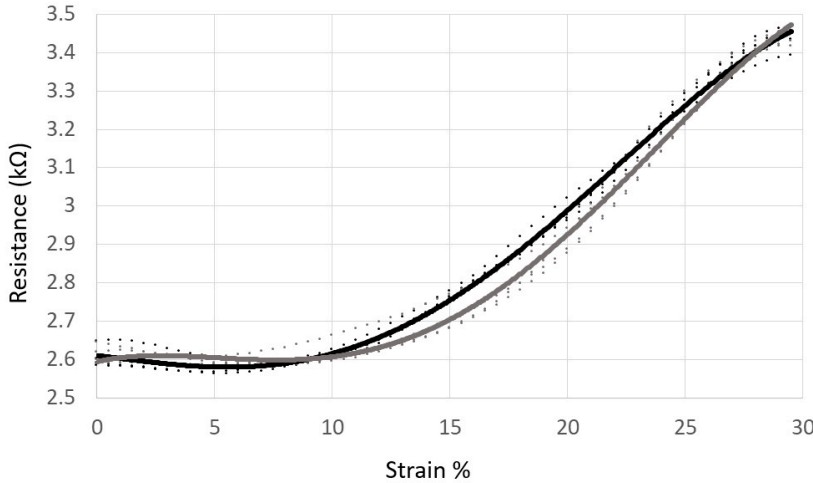

**Figure 2.** Conductive paint resistance vs strain graph, cycles 123–126 of 250 cycles. Dotted lines represent raw data with a 4th degree polynomial trendline for elongations (black line) and retractions (grey line).

For this study, we aimed to measure elbow and shoulder flexion. Therefore, one sensor was placed across each joint angle. For the elbow, the sensor was placed directly across the joint on the posterior

side and for the shoulder, just under the axilla, running parallel to the transverse plane towards the scapula (Figure 3).

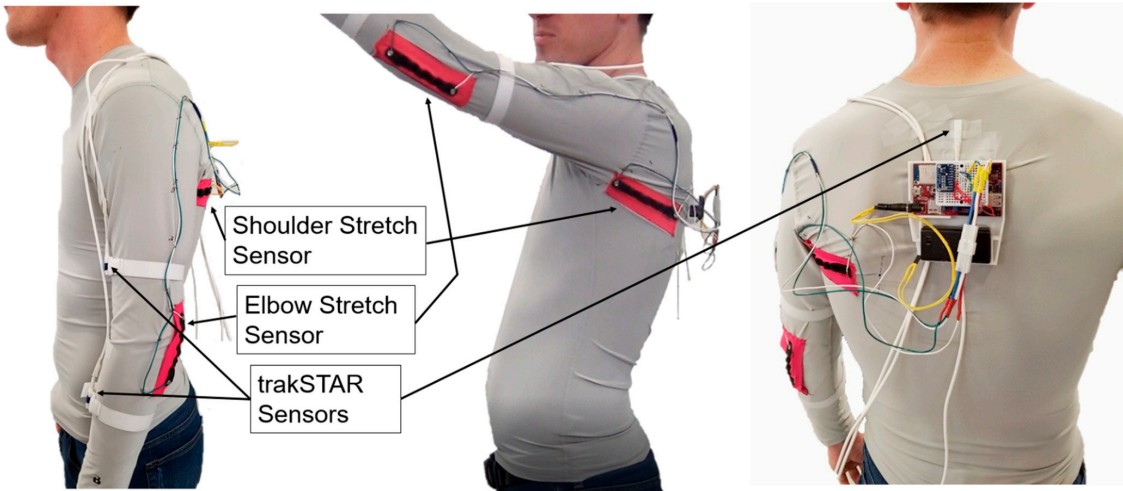

**Figure 3.** Sensor placements.

On both ends of each conductive paint sensor, a 5 mm tin-plated brass snap (Sewable Snaps—5 mm, Adafruit Industries, New York, NY, USA) was sewn into the sensor using conductive thread (Stainless Thin Conductive Thread—2 ply, Adafruit Industries, New York, NY, USA). The other side of the button snap was soldered to 22-gauge stranded copper wire that went from the sensor to the microcontroller worn on the users back. Sensor data were read by a Wi-Fi enabled microcontroller (WF32: Wi-Fi Enabled PIC32 Microcontroller, Digilent, USA). For improved accuracy, sensor data were read through a 16-Bit analog to digital converter (ADC) (ADS1115, Adafruit Industries, New York, NY, USA) rather than the 10-Bit on board ADC. A constant 5-volt signal was sent out by the microcontroller, and the voltage drop was recorded. Data were transmitted from the microcontroller through a router (Belkin N600 Router, Belkin, Playa Vista, CA, USA) over a Local Area Network (LAN) to a laptop with a custom written LabVIEW (LabVIEW, National Instruments, Austin, TX, USA) program at 18 Hz. Overall, the system aims to be comfortable by being wireless and untethered from a box, like the trakSTAR system, and incorporated within a comfortable, commercially available athletic garment. All aspects of the system from the garment through the microcontrollers are commercially available or open-sourced, so other researchers can replicate the methodology.

### 2.2. Garment Testing

For this study we tested two participants, one child (11-year-old male, 4′10′′ tall, 75 lbs) and one adult (24-year-old male, 6′1′′, 165 lbs) to evaluate the scalability of the design. The adult participant and parent of the child participant engaged in informed consent, while the child participant provided assent [IRB approval ([1110233-3), 08/30/2017). Each participant had a compression garment (Long-Sleeve Compression Tee, Badger, USA) made from an 83% polyester, 17% spandex blend purchased for their respective sizes (youth small and adult medium). Then the placement for the sensors was marked on the garment while the user was wearing it. After the fitting, each stretch sensor was stitched to the compression garment at both ends of the sensor, perpendicular to the direction of stretch in the locations described above.

Participants returned for testing in the laboratory when the garment was ready. Throughout all testing, participants wore the garment, as well as three IMUs from the trakSTAR (Ascension, St. Louis, MO, USA) motion analysis system to simultaneously track shoulder and elbow motion. A mid-range transmitter with Model 180, 6 degree of freedom sensors were used. One IMU was placed on the

anterior side of middle of the forearm, the next on the anterior side of the middle of the upper arm, and the last on the T3 vertebra (Figure 3).

### 2.3. Testing Protocol

Each participant went through three trials consisting of structured phases. Each trial consisted of a Calibration Phase and Static 1, Dynamic, Functional, and a second Static (Static 2) Phase. Data from Trial 1 of one of the participants for both the trakSTAR and conductive paint sensor smart garment from all phases can be seen in Figure 4.

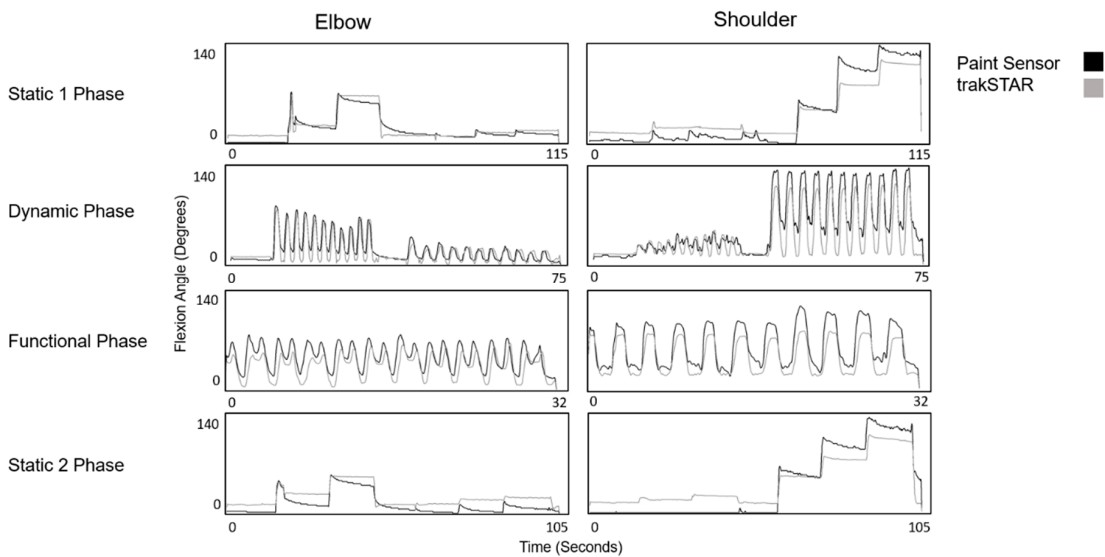

**Figure 4.** Graphs comparing data between the smart compression garment and trakSTAR from all phases for one trial.

### 2.3.1. Calibration

For all phases, the range of motion for the elbow was between 0 and 90 degrees, and for the shoulder was between 0 and 135 degrees. We selected to use a dynamic calibration where the participant moved their arm as they would in the dynamic assessment over 10 cycles of 0–90 degrees of flexion for the elbow and 10 cycles of 0–135 degrees of flexion for the shoulder. Then, the minimum and maximum values were recorded from both sensors. For the elbow, the minimum and maximum values were linearly mapped to 0 and 90 degrees, respectively, and for the shoulder, they were mapped to 0 to 135 degrees, respectively. These minimum and maximum values were kept constant for the remainder of the trial.

### 2.3.2. Static 1 Phase

The Static Phase consisted of a 10-s hold at distinct positions for both the elbow and the shoulder. One 10-s hold occurred at each of the following positions: 0°, 45°, and 90° of elbow flexion, and 0°, 45°, 90°, and 135° of shoulder flexion. Readings from the trakSTAR were used as the true reading.

### 2.3.3. Dynamic Phase

The Dynamic Phase consisted of moving the shoulder or elbow through a specific range of motion cyclically. The participant completed 10 cycles of elbow flexion between 0 and 90 degrees at 1 Hz then 10 cycles of shoulder flexion at 0.7 Hz. The different speeds were selected, so that the angular velocity participants moved at was similar. Participants coordinated their movement speed to a metronome.

### 2.3.4. Functional Phase

The Functional Phase incorporated a functional task that involved both elbow and shoulder flexion. Each participant repeatedly lifted an empty box from waist level to a shelf that was eye level. This task was repeated for 10 cycles at 0.7 Hz. Participants coordinated their actions to a metronome.

### 2.3.5. Static 2 Phase

The protocol for the Static 1 Phase was repeated a second time for a Static 2 Phase, so that reliability within the trial could be evaluated.

## 3. Results

Data were filtered through two, reversing-direction, second-order, low-pass Butterworth Filters with the cutoff frequency set at 3 Hz. For the Static Phase trials, the middle 6 s were selected from the 10-s hold for analysis. For Dynamic and Functional Phase trials, the middle 6 cycles of the 10 cycles were analyzed.

Descriptive analyses and the Intraclass Correlation Coefficient (ICC) were used to analyze the data. Using the trakSTAR system as the true angle, first the average absolute error between the two systems was found to identify the accuracy during individual study phases in the data collection. These results can be seen in Table 1(A) and were Static 1 Phase 9.98, Dynamic Phase 22.21, Functional Phase 25.02, and Static 2 Phase 16.49 degrees of error. Both Static Phases produced better results than the Dynamic and Functional Phases because in the Dynamic and Functional Phases, hysteresis did not allow the sensor to return to its original resistance before the next cycle began.

**Table 1.** Results across all trials.

| | **Average Absolute Error** | **Static 1** | | **Dynamic** | | **Functional** | | **Static 2** | |
|---|---|---|---|---|---|---|---|---|---|
| | | **Elbow** | **Shoulder** | **Elbow** | **Shoulder** | **Elbow** | **Shoulder** | **Elbow** | **Shoulder** |
| **A** | Participant 1 | 10.85 | 13.12 | 15.16 | 31.98 | 29.68 | 24.11 | 17.14 | 16.62 |
| | Participant 2 | 6.69 | 9.26 | 18.75 | 22.94 | 9.73 | 36.54 | 17.83 | 14.36 |
| | Participant Average | 8.77 | 11.19 | 16.96 | 27.46 | 19.71 | 30.33 | 17.49 | 15.49 |
| | Total Average | 9.98 | | 22.21 | | 25.02 | | 16.49 | |

| | **Average Absolute Error** | **Trial 1** | **Trial 2** | **Trial 3** |
|---|---|---|---|---|
| **B** | Participant 1 | 12.63 | 14.80 | 23.21 |
| | Participant 2 | 10.45 | 12.46 | 19.76 |
| | Total Average | 11.54 | 13.63 | 21.49 |

| | **Intraclass Correlation Coefficient** | **Dynamic** | | **Functional** | |
|---|---|---|---|---|---|
| | | **Elbow** | **Shoulder** | **Elbow** | **Shoulder** |
| **C** | Participant 1 | 0.949 | 0.960 | 0.800 | 0.933 |
| | Participant 2 | 0.847 | 0.892 | 0.749 | 0.901 |
| | Participant Average | 0.898 | 0.926 | 0.775 | 0.917 |
| | Total Average | 0.912 | | 0.846 | |

| | **Activity Counts** | **Dynamic** | | **Functional** | |
|---|---|---|---|---|---|
| | | **Elbow** | **Shoulder** | **Elbow** | **Shoulder** |
| **D** | Participant 1 | 96.97 | 81.08 | 87.10 | 75.00 |
| | Participant 2 | 98.42 | 69.57 | 61.29 | 77.78 |
| | Participant Average | 97.70 | 75.33 | 74.20 | 76.39 |
| | Total Average | 79.39 | | 75.00 | |

All data: *p*-value < 0.001.

Next the average absolute error between the two systems was taken across all phases within each of the 3 trials to assess reliability over time. These results can be seen in Table 1(B) and were Trial 1: 11.54, Trial 2: 13.63, and Trial 3: 21.49 degrees of error. These numbers show that the sensor decreased in accuracy over time, which is consistent with the Static 1 Phase producing better results than Static 2 Phase. A large portion of the increased error over the three trials came from the resistance exceeding the maximum resistance set during the calibration phase. With a more robust algorithm and calibration protocol, this error could be minimized. From averaging these results, it was also found that the elbow produced an average absolute error of 14.43 degrees and the shoulder 16.45 degrees. Averaging all the data resulted in an average absolute error of 15.55 degrees.

Despite the Dynamic and Functional Phases producing less accurate results than the Static Phases, the two systems were still highly correlated in these phases of movement. To show this, an Intraclass Correlation Coefficient for consistency was run for each trial of the Dynamic and Functional Phases. Significance was less than 0.001 for all tests. Averaging all the Dynamic Phase trials resulted in an ICC of 0.912, and averaging all of the Functional Phase trials resulted in an ICC of 0.846 (Table 1(C)). Despite the hysteresis of the system preventing a low mean error, there was an extremely strong correlation between the trakSTAR and the Smart Compression Garment.

Lastly, because the end goal was to see if the smart compression garment could record activity, not just kinematic data, the ability of the garment to record activity counts was also tested. We used the threshold method and defined an activity count as an angular velocity greater than 50 degrees per second. Angular velocity was filtered again, through two, reversing direction, 2nd order, low pass Butterworth Filters with the cutoff frequency set at 2 Hz. Activity counts were only analyzed during the Dynamic and Functional Phase trials because the Static Phase trials inherently lacked activity. The Dynamic Phase trials resulted in 79.39% accuracy and the Functional Phase trials resulted in 75.00% accuracy (Table 1(D)). While these accuracy values are already high, they would be expected to be significantly higher over a longer data collection that has more than 10 cycles for individual trials. There were roughly 10 activity counts registered over 10 cycles, thus not registering one activity count dropped the accuracy of the trial by 10%.

## 4. Discussion

In summary, there is a need for affordable, accessible, comfortable, and wireless forms of motion capture and activity classification that allows for data collections to be performed in natural settings during functional activities. Having a tool that meets these needs would allow researchers to have a more thorough understanding of movement and physical activity for a variety of patient populations outside of a lab setting. Building upon research that utilized conductive threads, we created a novel stretch sensor using conductive paint that changes in resistance when stretched. When this sensor was incorporated into a compression garment and placed across the shoulder and elbow joints, it accurately measured physical activity for both an adult and child, supporting the scalability of the solution. Activity counts were calculated throughout Dynamic Phase trials with 79.39% accuracy and throughout Functional Phase trials with 75.00% accuracy, with a combined average overall absolute error of 15.55 degrees compared to the trakSTAR data.

Accuracy of the smart compression garment can potentially be improved by a few factors. We decided on a dynamic calibration protocol where the minimum and maximum values during the specified range of motion were recorded. If the calibration protocol was expanded to account for the hysteresis during static tasks and resistance values not fully recovering during dynamic tasks, the results could potentially improve. In this study, we also only mapped resistance values from the stretch sensor to joint angle, but there would potentially be less hysteresis recorded if capacitance values were used instead. Having said this, the results clearly show a strong correlation during dynamic and functional tasks with the current testing protocol. A limitation of this study was that the compression values were not provided for the garments, and these data could not be evaluated in a timely manner due to lack of access to our textile testing facilities during the pandemic.

## 5. Conclusions

This prototype suggests stretch sensors made from commercially available conductive paint hold high potential for motion and activity tracking. Further research should focus on the integration of more sensors and the robustness of the system. If this technology will eventually be incorporated into a commercial product, additional research could include sealing the stretch sensors and utilizing hydrophobic coatings to make the garment machine washable. Researchers interested in replicating and modifying the research we have currently done can find a DIY manual with step by step instructions for how to fabricate the conductive paint stretch sensor [34]. While the system in this paper described only how to measure movement from the elbow and shoulder, this technology may be used to measure movement across many different joints. Additionally, individual movements can be mapped to control many different types of external stimuli as a new form of accessible interaction.

**Author Contributions:** Conceptualization, B.G. and M.A.L.; data curation, B.G.; formal analysis, B.G.; funding acquisition, M.A.L.; investigation, B.G. and M.A.L.; methodology, B.G. and M.A.L.; project administration, M.A.L.; resources, M.A.L.; software, B.G.; supervision, M.A.L.; validation, B.G.; visualization, B.G. and M.A.L.; writing—original draft, B.G.; writing—review & editing, M.A.L. All authors have read and agreed to the published version of the manuscript.

**Funding:** This work was supported by the National Science Foundation grant SCH: INT: Collaborative Research: Smart Wearable Systems to Support and Measure Movement in Children With and Without Mobility Impairments (#1722596).

**Conflicts of Interest:** The authors declare no conflict of interest.

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
