# Peer review of "Design and Initial Testing of an Affordable and Accessible Smart Compression Garment to Measure Physical Activity Using Conductive Paint Stretch Sensors"

_mti, doi:10.3390/mti4030045_

Round 1
Reviewer 1 Report
This work proposed a smart compression garment to measure physical activity, especially focus on range of motions (ROM) on elbows and shoulders.
Before the acceptance, there are several issues shoulder be tackled:
- The contribution of this work is not clear, which may limit the novelty of this work. The reviewer suggests that the more comparison between this work and previous works should be introduced.
- Please describe the specific design principles that enable this study to a more affordable, accessible, and comfortable motion capture systems.
- The protocol number of the IRB should be involved in the paper.
- The more subjects should be recruited. Two subjects are too less to support the reliability of this work. Also, any statistic analysis is meaningless if there is only two subjects...
- why did this work only focus on shoulder and elbow?
- The sampling rate of 2 Hz is too low to support dynamic activity tracking, especially for the upper limb motions.
Author Response
We thank the reviewers for their thoughtful suggestions for our manuscript. We are grateful for the opportunity to revise and resubmit the manuscript for a second review. Below we address each of the comments from the reviewers. Corresponding changes are marked within the manuscript in blue text.
Reviewer 1:
Comment: The contribution of this work is not clear, which may limit the novelty of this work. The reviewer suggests that the more comparison between this work and previous works should be introduced.
Response: We clarified that the contribution of this work was the conducting of pilot testing on the design and affordable and accessible smart compression garment. We used commercially available conductive paint, microcontrollers, and compression garments to reduce the cost and make the components more accessible to other researchers. This system is also wireless which is a large advantaged to wired IMU based systems. Current wireless systems are significantly more expensive than this system. Literature searches related to this topic reveal studies that have used conductive ink as part of a circuit in garments; no articles were found using conductive ink as sensors for movement. We added this to the introduction.
Comment: Please describe the specific design principles that enable this study to be more affordable, accessible, and comfortable motion capture systems.
Response: We added the cost of the conductive paint to highlight its affordability. We added that the paint is commercially available to highlight its accessibility. We clarified that the system is comfortable because it is wireless and untethered from a box like the trakSTAR system and is incorporated within a comfortable, commercially available athletic garment. All aspects of the system from the garment through the microcontrollers are commercially available or open sourced so other researchers can replicate the methodology. Other researchers can access detailed instructions to make the sensor by accessing the corresponding author’s web site.
Comment: The protocol number of the IRB should be included in the paper.
Response: We included this.
Comment: The more subjects should be recruited. Two subjects are too less to support the reliability of this work. Also, any statistic analysis is meaningless if there is only two subjects.
Response: We reworded the title and purpose to emphasize that this was a single-case study design testing with two participants of different ages and sizes to assess the initial accuracy and scalability of this design. The correlational statistic (ICC) performed is one commonly used to demonstrate agreement between two datasets (for example, to compare measurements of the same activity from different tools or to compare measurements of the same activity from the same tool but different raters). In the current manuscript, we used a single case study design with two participants in which multiple movement data points were collected using two different methods and then were statistically compared. The descriptive and correlational analyses are appropriate for the comparisons performed in this single case study design. (Onghena, 2005)
Comment: Why did this work only focus on shoulder and elbow?
Response: We clarified that we selected to measure elbow and shoulder flexion because movement at these joints is important for the performance of a variety of daily activities that are impaired for individuals with upper extremity disabilities. Measurement of movement at these joints is especially important to measure the effectiveness of exoskeletons and other devices that aim to improve arm function.
Comment: The sampling rate of 2 Hz is too low to support dynamic activity tracking, especially for the upper limb motions.
Response: We clarified that the sampling rate at which the sensor data was read into the software from the microcontroller was 18 Hz. This was the fastest sampling rate possible for the microcontroller while keeping a 16- bit resolution.
Additional Reference:
Onghena, P. (2005). Single case designs. In B. S. Everitt & D. C. Howell (Eds.), Encyclopedia of Statistics in Behavioral Science (Vol. 4, pp. 1850-1854). Chichester: John Wiley & Sons, Ltd.
Reviewer 2 Report
The first sentence of the abstract and introduction is exactly same. Para-phase any one.
Line 42, 68, 72, 200: avoid short form of words.
Section 2.1:
- Provide chemical composition of the conductive ink used.
- What measures were taken to ensure the fixation and binding of ink to the textile substrate?
- Provide toxicological information of the conductive ink used to establish its safety to human skin.
- How did you determine the ink you used will make a good bonding/adhesion with the textile materials you printed it in?
- How did you ensure the bonding was good enough?
Line 86 -90: breakdown the long sentence into shorter ones for better clarity.
Line 127: provide IRB reference number and approval date.
Section 2.2:
- Clarify if parental consent was taken for the child participant
- Provide a ranges of vital body measurements and weight for each participant,
- Provide corresponding garment measurements of the compression tees and garment sizes,
- Provide fabric specification (structure, weight, fibre compositions etc.) of the compression Tees used.
- Provide compression values of each garment
Line 195: mention the names of the statistical tests done.
Line 243- 246: Identify the reasons/factors of inaccuracy and discuss those. Does the flexibility of the textile material play any role in this case? If so, how?
Section 4:
- Have you tested fastness to rubbing and wash of the conductive ink? As the accuracy decreased gradually, it indicates, the fastness to rubbing is poor!
- You mentioned about hydrophobic treatments; there are a variety of them coming from different chemistries and application process. Which ones will match better with the conductive ink you used? How they will influence the conductivity/resistance of the sensor?
- Include a formal conclusion section
Author Response
Reviewer 2:
Comment: The first sentence of the abstract and introduction is exactly the same. Para-phrase any one.
Response: We reworded the first sentence of the introduction.
Comment: Line 42, 72, 200 – avoid short form of words.
Response: We corrected what we believe are the words the reviewer is referring to.
Section 2.1
Comment: Provide chemical composition of the conductive ink used.
Response: The chemical composition of the conductive isn’t available because, according to the manufacturer, it is a trade secret. The paint’s MSDS can be found here. The paint is comprised of water, natural resin, conductive carbon, and a humectant.
Comment: What measures were taken to ensure the fixation and binding of ink to the textile substrate? How did you determine the ink you used will make a good bonding/adhesion with the textile materials you printed it in? How did you ensure the bonding was good enough?
Response: We added text to further clarify how the paint was applied to the fabric and that the manufacturer of the paint lists that the paint can be applied to the selected fabric. The functional performance of the sensor demonstrated that the bonding was sufficient.
Comment: Line 86-90 – breakdown the long sentence into shorter ones for better clarity.
Response: We made this change.
Comment: Line 127 – provide IRB reference number and approval date.
Response: We added this information.
Section 2.2:
Comment: Clarify if parental consent was taken for the child participant.
Response: We clarified this was the case.
Comment: Provide a ranges of vital body measurements and weight for each participant.
Response: The height and weight of both participants have been added.
Comment: Provide corresponding garment measurements of the compression tees and garment sizes. Provide fabric specification (structure, weight, fibre compositions, etc.) of the compression tees used. Provide compression values of each garment.
Response: We added the garment sizes and fabric composition. Compression values were not provided for the garments and, due to Covid, we are not currently able to access the lab to perform additional testing.
Comment: Line 195 – mention the names of the statistical tests done.
Response: We added this information.
Comment: Line 243-246 – Identify the reasons/factors of inaccuracy and discuss those. Does the flexibility of the textile material play any role in this case? If so, how?
Response: We added this in the discussion section.
Section 4:
Comment: Provide toxicological information of the conductive ink used to establish its safety to human skin. Have you tested fastness to rubbing and wash of the conductive ink? As the accuracy decreased gradually, it indicates, the fastness rubbing is poor. You mentioned about hydrophobic treatments; there are a variety of them coming from different chemistries and application process. Which ones will match better with the conductive ink you used? How they will influence the conductivity/resistance of the sensor?
Response: The conductive paint is non-toxic but is not specifically cosmetically approved because that is not it’s intended use. The MSDS for the paint can be found at: https://www.bareconductive.com/wp-content/uploads/2020/05/2020.05.ElectricPaint-MSDS.pdf. We did not perform a fastness to rubbing test because the conductive paint wasn’t coated in the scope of this study. The conductive paint is non-toxic because it is water soluble. Because of this, the most effective hydrophobic treatment would be one that coats all sides of the fiber. Alternatively, a composite sensor could be created where the conductive paint layer is sandwiched between two insulative layers. This study aimed to identify if it was possible to create repeatable and reliable sensors from commercially available and inexpensive conductive paint. We added these suggestions from the reviewer as next steps for research.
Comment: Include a formal conclusion section.
Response: We added this.
Round 2
Reviewer 1 Report
The authors have tackled the proposed comments. At this moment, the paper could be accepted.
Author Response
We thank the reviewers for their thoughtful suggestions for our manuscript. We are grateful for the opportunity to revise and resubmit the manuscript for a second review. Below we address each of the comments from the reviewers. Corresponding changes are marked within the manuscript in blue text.
Reviewer 1:
Comment: The authors have tackled the proposed comments. At this moment, the paper could be accepted.
Response: We appreciate your time and constructive feedback to improve the manuscript.
Reviewer 2 Report
1) You said: "The chemical composition of the conductive isn’t available because, according to the manufacturer, it is a trade secret. The paint’s MSDS can be found here. The paint is comprised of water, natural resin, conductive carbon, and a humectant." Include this information in the manuscript.
2) You said: "...Compression values were not provided for the garments and, due to Covid, we are not currently able to access the lab to perform additional testing." Add this as a limitation of the study in the manuscript.
3) You said: "....The conductive paint is non-toxic because it is water soluble...". This is an unscientific claim!. Mention about the toxicological information of the paint in the manuscript citing from its MSDS.
Author Response
We thank the reviewers for their thoughtful suggestions for our manuscript. We are grateful for the opportunity to revise and resubmit the manuscript for a second review. Below we address each of the comments from the reviewers. Corresponding changes are marked within the manuscript in blue text.
Reviewer 2:
Comment: You said: "The chemical composition of the conductive isn’t available because, according to the manufacturer, it is a trade secret. The paint’s MSDS can be found here. The
paint is comprised of water, natural resin, conductive carbon, and a humectant." Include this information in the manuscript.
Response: We added this information in the manuscript.
Comment: You said: "...Compression values were not provided for the garments and, due to Covid, we are not currently able to access the lab to perform additional testing." Add this as
a limitation of the study in the manuscript.
Response: We added this limitation.
Comment: You said: "…The conductive paint is non-toxic because it is water soluble...". This is an unscientific claim. Mention about the toxicological information of the paint in the
manuscript citing from its MSDS.
Response: We added toxicological information from its MSDS.
We appreciate your time and constructive feedback to improve the manuscript.